# Unsupervised graph neural networks with recurrent features for solving combinatorial optimization problems

## Abstract

In recent years, graph neural networks (GNNs) have gained considerable attention as a promising approach to tackle combinatorial optimization problems. We introduce a novel algorithm, dubbed QRF-GNN in the following, that leverages the power of GNNs to efficiently solve combinatorial problems which have quadratic unconstrained binary optimization (QUBO) formulation. It relies on unsupervised learning and minimizes the loss function derived from QUBO relaxation. The key components of the architecture are the recurrent use of intermediate GNN predictions, parallel convolutional layers and combination of artificial node features as input. The performance of the algorithm has been evaluated on benchmark datasets for maximum cut and graph coloring problems. Results of experiments show that QRF-GNN surpasses existing graph neural network based approaches and is comparable to the state-of-the-art conventional heuristics.

## 1 Introduction

Combinatorial optimization (CO) is a well-known subject in computer science, bridging operations research, discrete mathematics and optimization. Informally, given some ground set, the CO problem is to select the combination of its elements, such that it lies on the problem's feasible domain and the cost of this combination is minimized. A significant amount of CO problems are known to be NP-hard, meaning that they are computationally intractable and the scope of application for exact algorithms to solve them is very narrow. Therefore, the development of heuristic methods that provide high-accuracy solutions in acceptable amount of time is a crucial challenge in the field Boussaïd et al. (2013).

A wide range of CO problems are defined on network data, and their solutions are encoded as a set of graph's edges or nodes. Such tasks are commonly met in many application domains such as logistics, scheduling etc. Implicit regularities and patterns often arise in graph structure and features, making the use of machine learning and especially graph neural networks (GNNs) very promising Cappart et al. (2023).

The quadratic unconstrained binary optimization (QUBO) is the problem to minimize a pseudo-Boolean polynomial $F(x)$ of degree two Boros & Hammer (1991):

$$\min_{x \in \{0,1\}^N} F(x), \quad F(x) = \sum_{i=1}^{N} \sum_{j=1}^{N} x_i Q_{ij} x_j = x^T Q x, \tag{1}$$

where $Q$ is the square matrix of coefficients. A huge number of CO problems can be formulated as QUBO Glover et al. (2022); Lucas (2014), which makes algorithms for its solution especially valuable in practice. Schuetz et al. (2022a) proposed to use the relaxed QUBO objective as a loss function to train unsupervised GNN. Such an approach allows using a graph network to solve a wide variety of problems formulated as QUBO with the same model. However, the ability of the proposed method to obtain accurate solutions has been widely debated in the community Boettcher (2023); Angelini & Ricci-Tersenghi (2023).

In this paper we introduce a novel unsupervised QUBO-based Graph Neural Network with a Recurrent Feature (QRF-GNN) for solving CO problems. We provide results of numerical experiments on popular benchmark instances of maximum cut (Max-Cut) and graph coloring problems. We show that the recurrent feature drastically improves the performance of GNNs owing to information about the current status of each node in the graph. A combination of artificial input features and parallel graph convolutional layers is offered for additional benefit. We show that the proposed method outperforms existing state-of-the-art (SOTA) approaches based on unsupervised learning for the considered problems. QRF-GNN is competitive with the best conventional heuristics in quality taking less time to find solutions for large graphs in some cases.

## 2 RELATED WORK

Graph neural networks are rapidly gaining popularity as a tools for solving CO problems Cappart et al. (2023). Supervised learning based approaches are common. A large number of models in this setting have been proposed to deal with travel salesman Prates et al. (2019); Joshi et al. (2022), graph coloring Lemos et al. (2019), maximum independent set (MIS) Li et al. (2018) etc. Supervised GNNs have also been successfully used for speeding up decision-making in conventional exact solvers, e.g. in branch-and-bound methods for mixed-integer programming. Two main directions here are imitation learning Gasse et al. (2019) framework and Neural Diving heuristics Nair et al. (2021). However, the need to collect labeled training instances into representative and unbiased dataset is a significant limitation of supervised algorithms.

One of the alternatives is to use reinforcement learning for iterative solution generation Mazyavkina et al. (2021). Khalil et al. (2017) proposed to use Deep Q-learning with a greedy node selection policy. The well-known REINFORCE algorithm has also been applied to learn greedy and local heuristics for travel-salesman and vehicle-routing Kool et al. (2019), SAT Yolcu & Poczos (2019) and other problems. Recently, the optimization of submodular Prajapat et al. (2023) and QUBO-based Rizvee & Khan (2023) reward functions has been investigated. While being promising approach, reinforcement learning methods may experience difficulties when facing large scale problems due to the vastness of the state space, and the need of a large number of samplings.

In our work we focus on unsupervised learning paradigm, meaning that GNN is trained to solve the particular problem instance end-to-end, optimizing its objective and aiming not to violate the constraints. Such methods, as well as in reinforcement learning, do not need a training set of already solved problems and can be considered as autonomous heuristic algorithms.

In Tönshoff J & M (2021) authors proposed RUN-CSP as a recurrent unsupervised neural network for the problems formulated in terms of maximum constraint satisfaction. The architecture includes a set of linear functions that provides message passing on a graph with nodes for all variables and edges for all constraints. After the message passing step current states as well as internal long-term states are updated by an LSTM cell. Based on the output, the network produces probabilities of a variable taking a certain value in the search domain. Amizadeh et al. (2018) came up with unsupervised GNN to solve SAT and CircuitSAT. They use directed acyclic graph representations of the problem and train the model that minimizes artificial loss function, whose minima correspond to a solution with a higher degree of satisfiability. Karalias & Loukas (2020) applied GNN in slightly different manner. It obtains a distribution of nodes corresponding to the candidate solution. The model is trained by minimizing the probabilistic penalty function, and sequential decoding is used to get the discrete solution, lowering the probability of its infeasibility. In recent work Wang et al. (2023) authors have introduced GNN-1N, adapting negative message passing technique into unsupervised GNN for solving graph coloring problem.

The use of continuous relaxation of the QUBO formulation for a particular problem as a loss function was suggested by Schuetz et al. (2022a) in their physics-inspired GNN (PI-GNN). The base architecture of PI-GNN consists of a trainable embedding layer to produce input features of nodes and several graph convolutional layers (GCN by Kipf & Welling (2017) or GraphSAGE by Hamilton et al. (2017b)). In Schuetz et al. (2022b) this model is applied for solving the graph coloring problem.

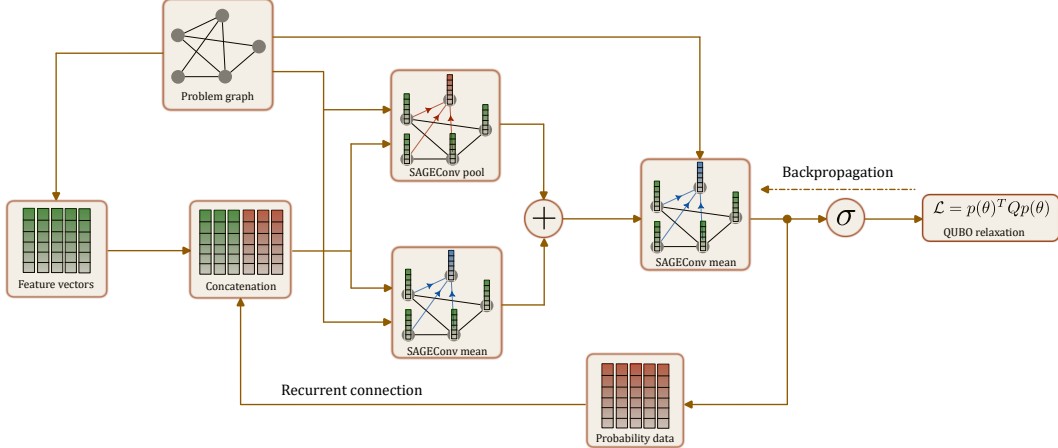

Figure 1: The QRF-GNN architecture. At each step the raw probability data from the previous step are concatenated with the artificial input feature vectors. Then these feature vectors along with the graph data pass through multiple SAGE layers according to the scheme to update probabilities.

## 3 QRF-GNN METHOD

Graph neural networks are capable to learn complex graph-structured data by capturing relational information. During training process, each of the nodes is associated with a vector which is updated based on the information from neighboring nodes. We consider an undirected graph $\mathcal{G} = (\mathcal{V}, \mathcal{E})$ with the vertex set $\mathcal{V} = \{1, \ldots, n\}$ and the edge set $\mathcal{E} = \{(i, j) : i, j \in \mathcal{V}\}$. Let $\boldsymbol{h}_i^l \in \mathbb{R}^{m_1}$ be the feature vector for the node $i$ and $\boldsymbol{h}_j^l \in \mathbb{R}^{m_1}$ the vector for the node $j$ at the $l$-th convolution step and let $e = (i, j)$ be the edge between nodes $i$ and $j$. We do not consider here the presence of a feature vector for edges.

To exchange information between nodes the message passing protocol is used. It can be divided into two main parts: message accumulation and message aggregation. Messages are accumulated using a general function $m_e^{l+1}$ which determines how information will be collected:

$$m_e^{l+1} = \phi\left(\boldsymbol{h}_i^l, \boldsymbol{h}_j^l\right), \quad (i, j) \in \mathcal{E}. \tag{2}$$

After that the feature vector for the node $i$ is updated by a function $\psi$ with a reduce or aggregate function $\rho$:

$$\boldsymbol{h}_i^{l+1} = \psi\left(\boldsymbol{h}_i^l, \rho\left(\{m_e^{l+1} : (i, j) \in \mathcal{E}\}\right)\right). \tag{3}$$

There are many different types of specific message-passing protocols that can be used Hamilton et al. (2017a). The architecture of QRF-GNN is based on the GraphSAGE framework Hamilton et al. (2017b). In this case the feature vector for each node $i$ aggregates information about a node's local neighborhood $\mathcal{N}(i)$ and receives an update according to the following scheme:

$$\begin{aligned} \boldsymbol{h}_{\mathcal{N}(i)}^{l+1} &= \rho\left(\{\boldsymbol{h}_j^l, \forall j \in \mathcal{N}(i)\}\right), \\ \boldsymbol{h}_i^{l+1} &= f\left(\boldsymbol{W}^l\left[\boldsymbol{h}_i^l \, \boldsymbol{h}_{\mathcal{N}(i)}^{l+1}\right]\right), \end{aligned} \tag{4}$$

where $f$ is a nonlinearity function and $\boldsymbol{W}^l$ is a matrix of trainable parameters acting on concatenated vectors $\boldsymbol{h}_i^l$ and $\boldsymbol{h}_{\mathcal{N}(i)}^{l+1}$. The size of the matrix $\boldsymbol{W}^l$ depends on the dimension of input vectors and the given size of hidden states. Typical aggregation methods $\rho$, such as averaging, pooling or LSTM-based aggregation Hamilton et al. (2017b), can be adopted here.

QRF-GNN uses three SAGE convolutions with different types of aggregation (see Figure 1). Mean and pool aggregation functions were chosen for two parallel intermediate SAGE layers, and the mean aggregation function was chosen for the last SAGE layer. This architecture configuration with a small number of successive layers allows to store more various information about local neighborhoods without much over-smoothing Rusch et al. (2023).

Being an unsupervised learning method, QRF-GNN optimizes the loss function which is essentially a relaxation of the QUBO problem. Replacing the binary decision variables $x_i \in \{0, 1\}$ with continuous probability parameters $p_i(\boldsymbol{\theta})$ yields:

$$
\begin{aligned}
\boldsymbol{x} \to \boldsymbol{p}(\boldsymbol{\theta}) &\in [0, 1]^{\times|\mathcal{V}|}, \\
F_{\text{QUBO}} = \boldsymbol{x}^T \boldsymbol{Q} \boldsymbol{x} \to \mathcal{L}_{\text{QUBO}}(\boldsymbol{\theta}) &= \boldsymbol{p}(\boldsymbol{\theta})^T \boldsymbol{Q} \boldsymbol{p}(\boldsymbol{\theta}),
\end{aligned}
\tag{5}
$$

where $\boldsymbol{\theta}$ are neural network parameters.

Now the trick is to recursively use the predicted probabilities data from the previous step as an additional feature of the node (see Figure 1) and directly utilize it through the message-passing protocol. It turns GNN training into an iterative optimization process in which the neural network adjusts the assignment of nodes classes under the influence of not only initial nodes properties, but also of the most probable classes of its neighbors. The use of recurrence makes a fundamental improvement in results over the incremental contribution from the architecture (see the ablation study in the Appendix).

---

**Algorithm 1:** Forward propagation of the QRF-GNN algorithm at the time step $t$

    **Input**  : Graph $\mathcal{G}(\mathcal{V}, \mathcal{E})$, input nodes features $\{\boldsymbol{a}_i, \forall i \in \mathcal{V}\}$
    **Output:** Probability $p_i$, hidden state $\boldsymbol{h}_i^t$,    $\forall i \in \mathcal{V}$

1   $\boldsymbol{h}_i^{t,0} \leftarrow \begin{bmatrix} \boldsymbol{a}_i \ \boldsymbol{h}_i^{t-1} \end{bmatrix}, \quad \forall i \in \mathcal{V}$;

2   **for** $i \in \mathcal{V}$ **do**

3      $\boldsymbol{h}_{\mathcal{N}(i)}^{t,1} \leftarrow \rho_{\text{mean}} \left( \left\{ \boldsymbol{h}_j^{t,0}, \forall j \in \mathcal{N}(i) \right\} \right)$;

4      $\boldsymbol{h}_i^{t,1} \leftarrow f \left( \boldsymbol{W}^1 \begin{bmatrix} \boldsymbol{h}_i^{t,0} \ \boldsymbol{h}_{\mathcal{N}(i)}^{t,1} \end{bmatrix} \right)$;

5      $\boldsymbol{h}_{\mathcal{N}(i)}^{t,2} \leftarrow \rho_{\text{pool}} \left( \left\{ \boldsymbol{h}_j^{t,0}, \forall j \in \mathcal{N}(i) \right\} \right)$;

6      $\boldsymbol{h}_i^{t,2} \leftarrow f \left( \boldsymbol{W}^2 \begin{bmatrix} \boldsymbol{h}_i^{t,0} \ \boldsymbol{h}_{\mathcal{N}(i)}^{t,2} \end{bmatrix} \right)$;

7   **end**

8   $\{\boldsymbol{h}_i^{t,1}, \forall i \in \mathcal{V}\} \leftarrow \text{BN}_{\gamma1,\beta1}(\{\boldsymbol{h}_i^{t,1}, \forall i \in \mathcal{V}\})$;

9   $\{\boldsymbol{h}_i^{t,2}, \forall i \in \mathcal{V}\} \leftarrow \text{BN}_{\gamma2,\beta2}(\{\boldsymbol{h}_i^{t,2}, \forall i \in \mathcal{V}\})$;

10   **for** $i \in \mathcal{V}$ **do**

11      $\boldsymbol{h}_i^{t,12} \leftarrow f(\boldsymbol{h}_i^{t,1} + \boldsymbol{h}_i^{t,2})$;

12      $\boldsymbol{h}_i^{t,12} \leftarrow \text{Dropout}(\boldsymbol{h}_i^{t,12})$;

13      $\boldsymbol{h}_{\mathcal{N}(i)}^{t,\text{out}} \leftarrow \rho_{\text{mean}} \left( \left\{ \boldsymbol{h}_j^{t,12}, \forall j \in \mathcal{N}(i) \right\} \right)$;

14      $\boldsymbol{h}_i^{t,\text{out}} \leftarrow f \left( \boldsymbol{W}^{\text{out}} \begin{bmatrix} \boldsymbol{h}_i^{t,12} \ \boldsymbol{h}_{\mathcal{N}(i)}^{t,\text{out}} \end{bmatrix} \right)$;

15   **end**

16   $p_i, \boldsymbol{h}_i^t \leftarrow \sigma(\boldsymbol{h}_i^{t,\text{out}}), \boldsymbol{h}_i^{t,\text{out}}, \quad \forall i \in \mathcal{V}$

---

The whole forward propagation process at the time step $t$ can be described in Algorithm 1. In Algorithm 1 the batch normalization (BN) has trainable parameters $\gamma$ and $\beta$ and is done over all nodes for each feature dimension.

There are several methods how to generate input features $\boldsymbol{a}_i$, which then go through the neural network. In this work, we create an artificial input feature vector as a composite vector of a random part, shared vector Cui et al. (2022) and pagerank Brin & Page (1998). At the first step, the probability vector $\boldsymbol{h}_i^0$ is initialized to zeros. Another way involves one-hot encoding for each node and then training a special embedding layer as in the PI-GNN Schuetz et al. (2022a) architecture. It allows the neural network itself to learn the most representative features. However, we did not use an embedding layer, since it requires additional computational resources and has shown no benefit over artificial features within the framework of conducted experiments (see the ablation study in the Appendix).

## 4 NUMERICAL EXPERIMENTS

In this paper, we considered two popular combinatorial optimization problems, namely Max-Cut and graph coloring. It is worth noting that our approach can potentially be applied to any problem that can be formulated as QUBO.

During experiments, we found out that there was no need to do a time-consuming optimization of hyperparameters (as it is done in the PI-GNN algorithm) to overcome existing GNNs. However, the hyperparameters tuning may lead to better results of QRF-GNN. The only parameters that we changed were the size of hidden layers and the maximum number of iterations, since for some types of graphs fewer iterations were sufficient to get a competitive result. We limited the number of iterations to $5 \times 10^4$ for random regular graphs and $10^5$ for all the other graphs, but in some cases convergence was reached much earlier. If the value of the loss function at the last 500 iterations had differed by less than $10^{-5}$ it was decided that the convergence was achieved and the training was stopped. In the case of the graph coloring problem, an additional stopping criterion was used and the solution was considered to be found when the absolute value of the loss function becomes less than $10^{-3}$. The learning rate was set empirically to 0.014 for the Adam optimizer, leaving the remaining parameters unaltered. The dropout was set to 0.5. The dimension of the random part of input vectors was equal to 10, the size of hidden layers was fixed at 50 for Max-Cut and at 140 for graph coloring.

Due to the stochasticity of the algorithm, it is preferable to do multiple runs with different seeds to find the best result. One can do separate runs in parallel possibly utilizing several GPUs. If the device has enough memory, the RUN-CSP scheme by Tönshoff J & M (2021) can be used. In this case, one composite graph with duplicates of the original one is created for the input. We trained the model in parallel on the NVIDIA Tesla V100 GPU. Conventional heuristics were launched on the machine with two Intel Xeon E5-2670 v3 @ 2.30GHz. The computational time depended on the size of the graph and the achievement of convergence, but the longest run of QRF-GNN did not exceed 17 minutes.

### 4.1 MAX-CUT

Max-Cut is a classic well-known combinatorial problem, which is to divide the vertices of the graph $\mathcal{G} = (\mathcal{V}, \mathcal{E})$ into two subsets such that the number of edges whose adjacent nodes are in different subsets is maximum (or the total weight of such edges in the case of a weighted graph). Its QUBO formulation is as follows:

$$\max \quad \frac{1}{2} \sum_{i<j} w_{ij}(1 - x_i x_j),$$
$$\text{s.t.} \quad x_i \in \{0, 1\}, \quad \forall i \in \mathcal{V}. \tag{6}$$

where $w_{ij}$ is the weight of the edge $(i, j)$ in the graph $\mathcal{G}$. In this paper we consider only unweighted graphs with $w_{ij} = 1, \forall (i, j) \in \mathcal{E}$.

The objective function 6 can be rewritten as follows:

$$F_{\text{QUBO}} = \mathbf{x}^T Q \mathbf{x} = \sum_{i,j} x_i Q_{ij} x_j, \quad Q = A - D, \tag{7}$$

where $A$ is an adjacency matrix of the graph and $D$ is a diagonal matrix with degrees of corresponding nodes.

To evaluate the performance of QRF-GNN, we compared results with other GNNs and popular heuristics on several datasets.

Table 1 shows the mean *P-value* $= \sqrt{\frac{4}{d}} \left( \frac{z}{n} - \frac{d}{4} \right)$ obtained with extremal optimization heuristics (EO) by Boettcher (2003), RUN-CSP and QRF-GNN for random regular graphs with $n = 500$ nodes and different degrees $d$. Here $z$ corresponds to the obtained cut size. Results of EO and

Table 1: *P-value* of EO, PI-GNN, RUN-CSP amd QRF-GNN for d-regular graphs with 500 nodes and different degree $d$. The best graph cuts are in bold. Values of PI-GNN with GCN or SAGE layers are written through a slash (gaps mean data are not published). The results of PI-GNN on graphs with $d = 3$ are given for the GCN and SAGE convolutional layers. The results of QRF-GNN on graphs with $d = 3$ are given for the best out of 5 and 15 runs.

| d | EO | PI-GNN | RUN-CSP | QRF-GNN |
|---|------|-------------|---------|-------------|
| 3 | **0.727** | 0.612/0.678 | 0.714 | 0.725/**0.727** |
| 5 | 0.737 | 0.608 | 0.726 | **0.738** |
| 10 | 0.735 | - | 0.710 | **0.737** |
| 15 | 0.736 | - | 0.697 | **0.739** |
| 20 | 0.732 | - | 0.685 | **0.735** |

Table 2: Number of cuts for benchmark instances from Gset Ye (2003) with the number of nodes $|\mathcal{V}|$ and the number of edges $|\mathcal{E}|$. The best results are in bold.

| Graph | $|\mathcal{V}|$ | $|\mathcal{E}|$ | BLS | TSHEA | EO | PI-GNN | RUN-CSP | QRF-GNN |
|-------|------|-------|---------|---------|-------|--------|---------|---------|
| G14 | 800 | 4694 | **3064** | **3064** | 3058 | 3026 | 2943 | 3058 |
| G15 | 800 | 4661 | **3050** | **3050** | 3046 | 2990 | 2928 | 3049 |
| G22 | 2000 | 19990 | **13359** | **13359** | 13323 | 13181 | 13028 | 13340 |
| G49 | 3000 | 6000 | **6000** | **6000** | **6000** | 5918 | **6000** | **6000** |
| G50 | 3000 | 6000 | **5880** | **5880** | 5878 | 5820 | **5880** | **5880** |
| G55 | 5000 | 12468 | 10294 | **10299** | 10212 | 10138 | 10116 | 10282 |
| G70 | 10000 | 9999 | 9541 | 9548 | 9433 | 9421 | 9319 | **9559** |

RUN-CSP were taken from Tönshoff J & M (2021), where *P-value* was averaged over 1000 graphs. RUN-CSP was allowed to make 64 runs for each graph and in the case of EO the best of two runs was chosen Yao et al. (2019). *P-values* of PI-GNN depend on the particular architecture. Results for graphs with a degree 3 and 5 were published in Schuetz et al. (2022a) for the architecture with GCN layer, and it corresponds to the first value in the column for PI-GNN. The cut size was bootstrap-averaged over 20 random graph instances and PI-GNN took up to 5 shots. In the paper by Schuetz et al. (2023) the authors considered another option with the SAGE layer and showed that in this case the results for graphs with a degree 3 can be improved by 10.78%. This is reflected in the second value in the corresponding column. Results for graphs with other $d$ are omitted since they were not published.

For QRF-GNN we estimated the *P-value* on two sets of 100 graphs and noticed the negligible difference. Therefore, the final *P-value* in the table is obtained by averaging over 200 graphs. We present results for the best value out of 5 and 15 runs for graphs with $d = 3$ and the best out of 5 runs for the rest. As one can see, QRF-GNN clearly outperforms RUN-CSP and PI-GNN in all cases and starting from $d = 5$ shows the best results over all considered algorithms. Increasing the number of runs to 15 allows QRF-GNN to show the best result on graphs with $d = 3$ as well.

We also evaluated QRF-GNN on benchmark graphs from a Gset collection Ye (2003). To make the evaluation more informative, we implemented $\tau$-EO heuristic from Boettcher & Percus (2001). As suggested by authors, we set $\tau = 1.3$ and the number of single spin updates was limited by $10^7$. For small graphs $\tau$-EO can find a high-quality solution, but with increase of the graph size the accuracy of the algorithm degrades due to the limited number of updates. This behavior is expected by the authors, who suggested optimal scaling for number of updates as $\sim O(|\mathcal{V}|^3)$. However, it is computationally expensive to carry out the required number of iterations. We performed 20 runs of EO with different initializations to partially compensate for this. Within the given limit, the EO algorithm took $\sim 6800$ seconds per run to obtain a solution for the relatively large graph G70.

Table 2 contains the comparison results of our algorithm with PI-GNN, RUN-CSP, EO and the SOTA heuristics Breakout Local Search (BLS) Benlic & Hao (2013) and the Tabu Search based Hybrid Evolutionary Algorithm (TSHEA) Wu et al. (2015). QRF-GNN as well as BLS and TSHEA

Table 3: Number of conflicts for the given chromatic number $\chi$ achieved by HybridEA, GNN-1N, PI-GNN, GDN and QRF-GNN on graphs from the COLOR dataset Trick (2002). Gaps in columns for GNN-1N, GDN and PI-GNN mean that the data has not been published.

| Graph | $|\mathcal{V}|$ | $|\mathcal{E}|$ | $\chi$ | HybridEA | GNN-1N | PI-GNN | GDN | QRF-GNN |
|---|---|---|---|---|---|---|---|---|
| anna | 138 | 493 | 11 | 0 | - | 0 | 0 | 0 |
| david | 87 | 406 | 11 | 0 | - | - | 0 | 0 |
| games120 | 120 | 638 | 9 | 0 | - | - | 0 | 0 |
| homer | 561 | 1629 | 13 | 0 | - | - | 0 | 0 |
| huck | 74 | 301 | 11 | 0 | - | - | 0 | 0 |
| jean | 80 | 254 | 10 | 0 | - | 0 | 0 | 0 |
| myciel5 | 47 | 236 | 6 | 0 | 0 | 0 | 0 | 0 |
| myciel6 | 95 | 755 | 7 | 0 | 0 | 0 | 0 | 0 |
| queen5-5 | 25 | 160 | 5 | 0 | 0 | 0 | 0 | 0 |
| queen6-6 | 36 | 290 | 7 | 0 | 0 | 0 | 0 | 0 |
| queen7-7 | 49 | 476 | 7 | 0 | 0 | 0 | 9 | 0 |
| queen8-8 | 64 | 728 | 9 | 0 | 1 | 1 | - | 0 |
| queen9-9 | 81 | 1056 | 10 | 0 | 1 | 1 | - | 0 |
| queen8-12 | 96 | 1368 | 12 | 0 | 0 | 0 | 0 | 0 |
| queen11-11 | 121 | 1980 | 11 | 14 | 13 | 17 | 21 | 7 |
| queen13-13 | 169 | 3328 | 13 | 18 | 15 | 26 | 33 | 15 |
| Cora | 2708 | 5429 | 5 | 0 | - | 0 | 0 | 0 |
| Citeseer | 3327 | 4732 | 6 | 0 | - | 0 | 0 | 0 |
| Pubmed | 19717 | 44338 | 8 | 0 | - | 17 | 21 | 0 |

was run 20 times on each graph. The best attempt out of 64 was chosen for RUN-CSP in original papers. In order to obtain the results of PI-GNN, the authors applied hyperparameter optimization for each graph. The results of RUN-CSP for the G70 graph was obtained by running the code[1] with parameters reported in Tönshoff J & M (2021).

Our algorithm outperforms both neural network approaches, but it is inferior to the SOTA heuristics. It gives the same results on graphs G49 and G50 and provides better results only for the G70 graph. It should be noted that the solution search time for the G70 graph was 1033 s in the case of QRF-GNN, while BLS and TSHEA took several times more, or about 11365 s and 6932 s correspondingly Benlic & Hao (2013); Wu et al. (2015).

## 4.2 COLORING

We distinguished two formulations in the graph coloring problem.

The first formulation requires to color the graph $\mathcal{G} = (\mathcal{V}, \mathcal{E})$ with a given number of colors without violations. The second formulation is to find the minimum number of colors in which the graph can be colored by the algorithm. Here, coloring a graph means assigning a certain color to each vertex, and violation occurs when adjacent vertices have the same color.

The QUBO model is formulated as follows:

$$
\begin{aligned}
\min \quad & \sum_i \left(1 - \sum_c x_{i,c}\right)^2 + \sum_{(i,j)\in} \sum_c x_{i,c} x_{j,c}, \\
\text{s.t.} \quad & x_{i,c} \in \{0,1\}, \quad \forall i \in \mathcal{V}, \quad \forall c \in \{1,\dots,k\},
\end{aligned}
\tag{8}
$$

where $k$ is the number of colors the graph has to be colored.

---

[1] https://github.com/toenshoff/RUN-CSP

Table 4: The number of color needed for coloring without conflicts by HybridEA, PI-GNN, RUN-CSP and QRF-GNN. Gaps in columns for RUN-CSP and PI-GNN mean that the data has not been published. Results of PI-GNN were obtained by applying randomized post-processing heuristic to the solutions with conflicts Schuetz et al. (2022b).

| Graph | $\chi$ | HybridEA | PI-GNN | RUN-CSP | QRF-GNN |
|---|---|---|---|---|---|
| anna | 11 | 11 | 11 | 11 | 11 |
| david | 11 | 11 | - | 11 | 11 |
| games120 | 9 | 9 | - | 9 | 9 |
| homer | 13 | 13 | - | 17 | 13 |
| huck | 11 | 11 | - | 11 | 11 |
| jean | 10 | 10 | 10 | 10 | 10 |
| myciel5 | 6 | 6 | 6 | 6 | 6 |
| myciel6 | 7 | 7 | 7 | 8 | 7 |
| queen5-5 | 5 | 5 | 5 | 5 | 5 |
| queen6-6 | 7 | 7 | 7 | 8 | 7 |
| queen7-7 | 7 | 7 | 7 | 10 | 7 |
| queen8-8 | 9 | 9 | 10 | 11 | 9 |
| queen9-9 | 10 | 10 | 11 | 17 | 10 |
| queen8-12 | 12 | 12 | 12 | 17 | 12 |
| queen11-11 | 11 | 12 | 14 | >17 | 12 |
| queen13-13 | 13 | 14 | 17 | >17 | 15 |
| Cora | 5 | 5 | 5 | - | 5 |
| Citeseer | 6 | 6 | 6 | - | 6 |
| Pubmed | 8 | 8 | 9 | - | 8 |

Table 5: Approximate runtime in seconds for PI-GNN and QRF-GNN training on a single GPU on instances from the COLOR dataset and citation graphs.

| Graph | $|\mathcal{V}|$ | $|\mathcal{E}|$ | PI-GNN, $\times 10^3 s$ | QRF-GNN, $\times 10^3 s$ |
|---|---|---|---|---|
| COLOR | 25-561 | 160-3328 | 3.6 ÷ 28.8 | 0.002 ÷ 1 |
| Cora | 2708 | 5429 | 0.3 | 0.06 |
| Citeseer | 3327 | 4732 | 2.4 | 0.018 |
| Pubmed | 19717 | 44338 | 24 | 0.156 |

The loss function can be reduced to the second term of the objective 8 in order to train QRF-GNN or PI-GNN. The condition for the uniqueness of the color assigned to the vertex which the first term specifies is met automatically by the softmax layer.

We performed experiments on graphs from the COLOR dataset Trick (2002) and additional calculations for three citation graphs Cora, Citeseer and Pubmed Li et al. (2022). Results of graph coloring with the optimal number of colors are compared with other neural network architectures, such as PI-GNN Schuetz et al. (2022b), GNN-1N Wang et al. (2023) and GDN Li et al. (2022), and the SOTA heuristics HybridEA Galinier & Hao (1999) (see Table 3). Missing results for PI-GNN mean that they are not contained in the source paper. The code[2] which was used for the HybridEA estimation is based on Lewis (2021). To evaluate the results of QRF-GNN, we did up to 10 runs for some graphs, although most of them required only one run. It can be seen from Table 3 that QRF-GNN shows the best results on all instances.

---

[2] http://rhydlewis.eu/gcol/

Table 4 shows the number of colors that the algorithm needs to color the graph without violations. Results of GNN-1N and GDN were not shown in the original paper. We also added results of RUN-CSP taken from Tönshoff J & M (2021).

Blanked lines in Table 4 refer to that results have not been published. To find the number of colors required to color the graph without violations, we successively increased the number of colors in each new run until the correct coloring was found among 10 seeds. It is clear from Table 4 that QRF-GNN is superior to the presented neural network based algorithms on all considered graphs. It is comparable to the SOTA heuristics HybridEA, and it shows the worse result only on 1 graph out of 19. The HybridEA method works much faster on considered graphs and converges within 1 ms to 6.5 s, however, this difference becomes smaller as the graph size increases. The convergence time for PI-GNN and QRF-GNN is shown in Table 5. The number of epochs for convergence of QRF-GNN on citation graphs was no more than 6000 and varied for the COLOR dataset from $\sim$200 to $9 \times 10^4$. The estimated runtime for QRF-GNN turned out to be significantly less than for PI-GNN, and in some cases the difference reaches more than three orders of magnitude. This is due to the fact that QRF-GNN does not require exhaustive tuning of hyperparameters for each instance in contrast to PI-GNN Schuetz et al. (2022b).

## 5 DISCUSSION AND CONCLUSIONS

In this work, we present a novel graph neural network architecture called QRF-GNN, which is able to solve CO problems in unsupervised mode. The training process is performed by using the continuous relaxation of the QUBO formulation as a loss function, thus the neural network predicts the probability that each decision variable takes one of the binary values. The key element of this architecture is the recurrent feature, which provides information about the probability of belonging to a certain class for neighboring nodes to GNN at each iteration. Using of the mixture of parallel convolutions with different aggregation functions and special artificial input features further improves the overall performance.

The ability of unsupervised GNNs, and in particular those using the relaxed QUBO as a loss function, to accurately solve CO problems has been debated in the scientific community. Responding to Schuetz et al. (2022a), Boettcher (2023) had made a comparative analysis for the Max-Cut on 3-regular graphs and concluded that GNN based approach can not outperform greedy algorithms and provided solutions much worse than the reputable EO Boettcher & Percus (2001) heuristics. In Angelini & Ricci-Tersenghi (2023) authors also claim that modern GNNs do worse than classical greedy algorithms in solving CO problems, providing experiments on random instances for MIS.

We provide comparative analysis showing that our approach is superior to existing GNNs in solving Max-Cut and graph coloring problems. QRF-GNN is able to outperform conventional EO heuristic on d-regular random graphs and instances from the Gset dataset. It is competitive to SOTA heuritics like BLS and TSHEA for Max-Cut as well as to HybridEA for graph coloring. We consider this result to be significant, taking into account that most of the mentioned methods are specified to solve particular problems, while QRF-GNN is potentially able to solve any CO problem that can be mapped to QUBO and demonstrate better performance on large instances.

## 6 REPRODUCIBILITY STATEMENT

To ensure that the results of the article can be reproduced, we have provided the scheme of the QRF-GNN architecture and its pseudocode in section 3. A description of the main parameters and convergence criteria is contained in section 4. We have also included additional training details and information about utilized packages in Appendix. It is worth noting that due to the stochastic behavior of the algorithm the results may differ depending on the initialization. We encourage everyone to collect enough statistics by doing several independent runs to obtain high-quality results.

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

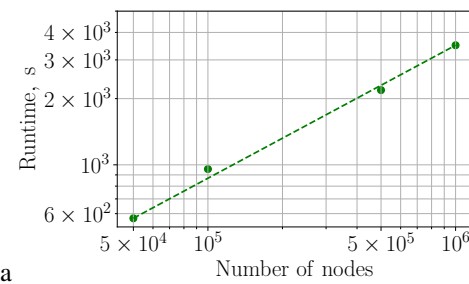 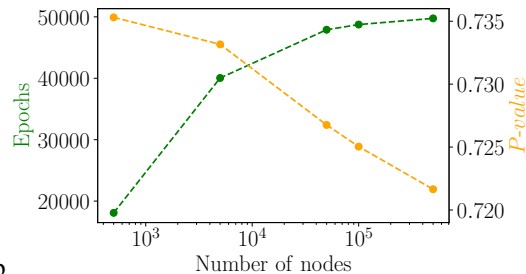

a                                                           b

Figure 2: a) The computation time of $5 \times 10^4$ epochs of QRF-GNN on random regular graphs with $d = 5$ in the sparse format depending on the number of nodes. b) The epoch number averaged over 20 graphs at which the algorithm found the best solution during the training process (green) and the mean *P-value* for 1 run (orange).

## A    TECHNICAL DETAILS AND CONVERGENCE

All random graphs in this work were generated by the NetworkX[3] package. To implement the QRF-GNN architecture the DGL library[4] was used. We did not define a learning rate schedule for the optimizer. Gradients were clipped at values 2 of the Euclidean norm.

We tested three ways to recursively utilize the probability data. Specifically, we passed raw probability data taken before the sigmoid layer, data after the sigmoid layer or concatenated both of these options. Different recurrent features led to a minor improvement on some graphs, while at the same time slightly worsening the results on other graphs. In this work we presented results for the concatenated data.

The number of runs and iterations in experiments were not optimal and were chosen for a more fair comparison with other algorithms. More runs and iterations can lead to better results. We conducted additional experiments on 20 random regular graphs with $d = 5$ and up to one million nodes. One run was made for each graph and the number of iterations was limited to $5 \times 10^4$. The training time for large graphs in sparse format on single GPU is shown in Figure 2a. We also analyzed how the number of iterations can affect the quality of the solution. As the number of vertices increases, the epoch number at which the last found best solution was saved moves closer to the specified boundary (see Figure 2b). Meanwhile, the average *P-value* of one run drops from 0.735 to 0.722 and one of the reasons for this may include the limited duration of training. If, for example, we train QRF-GNN for $10^5$ epochs on graphs with $n = 5 \times 10^4$ nodes, the average *P-value* will increase from 0.726 to 0.728, while on small graphs with $n = 500$ we do not observe such an effect. Thus, it is difficult to talk about the convergence of the algorithm on large instances under the given constraint. The recommendation is to follow the latest best solution updates and terminate the algorithm if it does not change for a sufficiently large ($> 10^4$) number of epochs.

In order to study the robustness of the algorithm with respect to changes in hyperparameters, we run the default QRF-GNN architecture with two parallel layers on graphs from the Gset dataset for the Max-Cut problem. All hyperparameters except the learning rate were chosen as described in Section 4. As can be seen from Figure 3, small values of the learning rate do not allow to achieve convergence in $10^5$ iterations. For the learning rate greater than 0.01 the results become relatively stable and the best number of cuts is achieved for values from 0.01 to 0.02.

## B    ABLATIONS OF QRF-GNN

We analyzed which components of QRF-GNN make the greatest contribution to its performance on the example of Max-Cut problem-solving. The default architecture includes two intermediate SAGEConv layers and the recurrent feature. The most dramatic drop in quality occurs if the recurrent part is excluded (see Figure 4).

---

[3] https://networkx.org/
[4] https://www.dgl.ai/

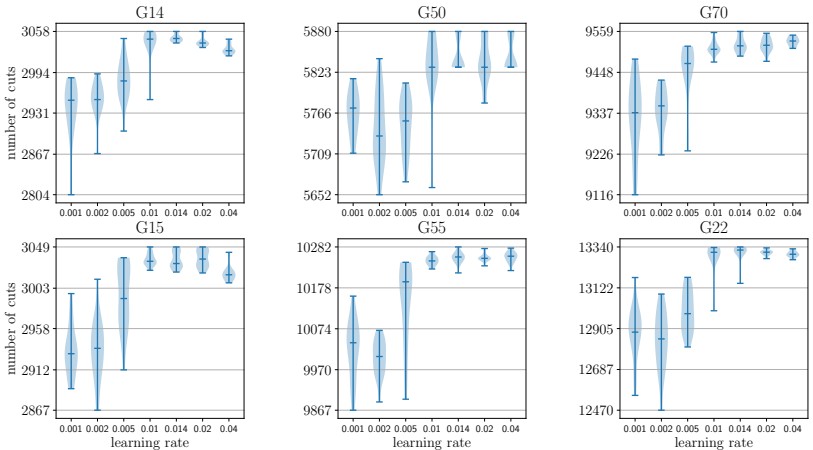

Figure 3: Results distribution for 20 runs of the default QRF-GNN architecture depending on the learning rate for the Max-Cut problem on several instances from Gset. The number of iterations in all cases was fixed at $10^5$. Horizontal lines mean maximum, median and minimum values.

Table 6: The first row contains the average *P-value* over 200 random regular graphs with $d = 5$ for 1 run of QRF-GNN with same configurations as in Figures 5 and 6. The second row shows the average *P-value* over 200 graphs when the best cut out of 5 runs is taken.

| Runs | Default | no SAGEConv (mean) | no SAGEConv (pool) |
|------|---------|--------------------|--------------------|
| 1    | 0.734   | 0.732              | 0.725              |
| 5    | 0.738   | 0.737              | 0.734              |

Throwing out one of the intermediate convolutional layer does not result in such a strong downgrade (see Figure 5). However, the absence of the convolutional layer with a pool aggregation function leads to a decrease in the median result, upper bound, and an increase in the results dispersion for almost all graphs. Discarding the layer with a mean aggregation function can increase the median cut and even decrease the variance in some cases, but upper bounds either stay the same or decrease, even if we double the number of parameters in the remaining hidden layer (see Figure 5). Further ablation on random regular graphs shows that the absence of any convolutional layer leads to a worse result(see Figure 6). Table 6 with average results for one seed and the best of the five seeds also confirms the advantage of using a combination of two layers.

In this work, we settled on combining a random vector, shared vector Cui et al. (2022) and pagerank Brin & Page (1998) for the input feature vector by default. Fig. 7 shows the results when one of the parts (random vectors or the pagerank of nodes) was removed. In some cases, the median improves after dropping features, but the upper bound tends to only go down as the number of input features decreases. Using an embedding layer does not show any benefit over the default version.

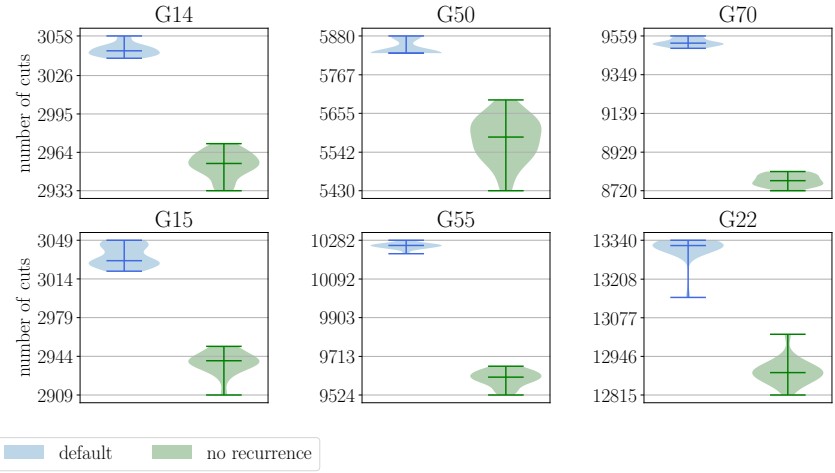

Figure 4: Results distribution for 20 runs of QRF-GNN with (blue) and without (green) the recurrent connection on several instances from Gset. Horizontal lines mean maximum, median and minimum values.

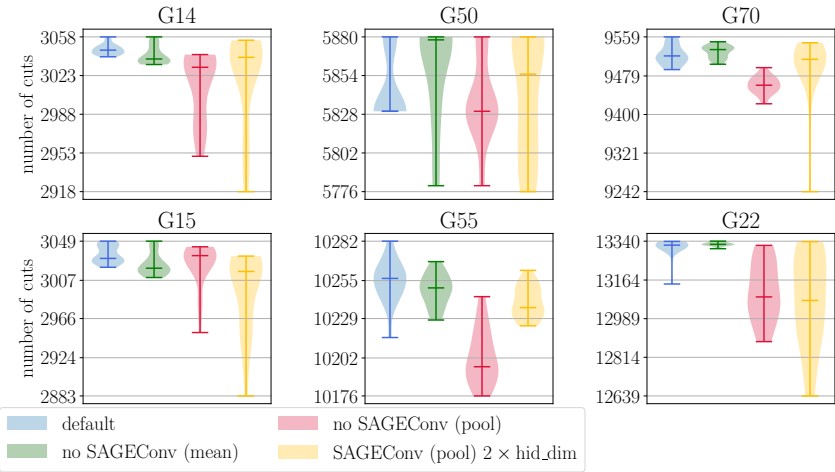

Figure 5: Results distribution for 20 runs of QRF-GNN of the default architecture (blue), with the absence of one SAGE layer with a mean aggregation function (green) or the SAGE layer with a pool aggregation function (red). Horizontal lines mean maximum, median and minimum values.

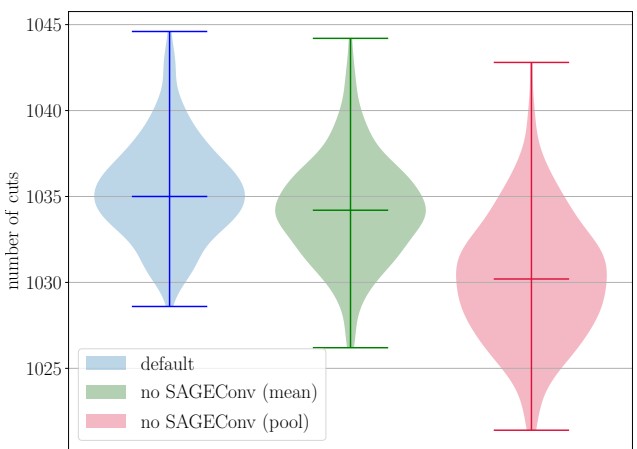

Figure 6: Results distribution for the mean of 5 runs on 200 random regular graphs with $d = 5$ and 500 nodes. QRF-GNN had the same configurations as in the Fig. 5.

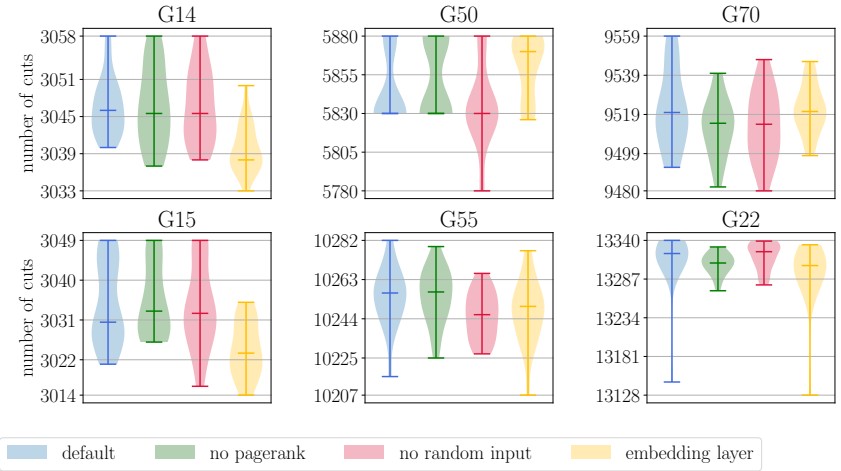

Figure 7: Results distribution for 20 runs of QRF-GNN with the default architecture on several instances from Gset. Input feature vectors varied as follows: the default choice corresponds to the blue color; the exclusion of the pagerank component corresponds to the green color; the exclusion of the random part corresponds to the red color. The use of a trainable embedding layer instead of artificial features is indicated in yellow.

