# OpenReview forum: "Unsupervised graph neural networks with recurrent features for solving combinatorial optimization problems"
_ICLR.cc/2024/Conference — Submitted to ICLR 2024_

### Official Review · Reviewer_vpKg · 2023-10-24

**Soundness:** 3 good
**Presentation:** 3 good
**Contribution:** 2 fair
**Rating:** 5
**Confidence:** 2

**Summary:**

The paper discusses the emergence of graph neural networks (GNNs) as a solution for combinatorial optimization problems. The authors introduce a new algorithm called QRF-GNN, which employs GNNs to efficiently address problems with quadratic unconstrained binary optimization (QUBO) formulations. QRF-GNN employs unsupervised learning and minimizes a loss function derived from QUBO relaxation. Its architecture includes recurrent use of intermediate GNN predictions, parallel convolutional layers, and a combination of artificial node features as input.

**Strengths:**

1. The paper is well written and well presented.

2. The paper also focuses on an important problem QUBO problems via unsupervised learning (which may have certain benefits)

3. The paper shows experimental evidence that suggests that their method indeed works well.

**Weaknesses:**

Please refer to questions.

**Questions:**

1. I'm uncertain about the primary contributions of the paper. Are the key contributions limited to the QRF-GNN architecture and the incorporation of random node features combined with pagerank? Or do they also encompass the formulation of the problem as an unsupervised learning approach?

2. If the primary contributions are solely related to the architecture and node features, it's worth noting that while the results appear promising, the method might be viewed as somewhat incremental, especially when compared to existing methods like spectral clustering, which can be considered a basic form of unsupervised learning. In such a case, could you provide insights into any theoretical guarantees, if any exist for your work?

3. I have concerns about the reproducibility of this work, as it appears the authors haven't made the code base available. Additionally, they mention that the method lacks stability. How do you determine the best run under these circumstances?

As it stands, I am inclined to recommend rejecting this paper. However, since I am not an expert in this field, I am open to revising my evaluation if the authors address these questions or if other reviewers support the work.

---

> ### Author Response · Authors · 2023-11-15
> **Response to Reviewer vpKg**
>
> We thank the reviewer vpKg for their comment and for showing interest to our work.
>
> $ \\ $
>
> #### •  **Q1. I'm uncertain about the primary contributions of the paper. Are the key contributions limited to the QRF-GNN architecture and the incorporation of random node features combined with pagerank? Or do they also encompass the formulation of the problem as an unsupervised learning approach?**
>
> The main contribution of the paper is that the use of recurrent features drastically improves the performance of GNNs in solving Combinatorial Optimization (CO) problems. As a result, GNN obtains highly accurate solutions and can even compete  with the best heuristic algorithms for solving MAX-CUT and graph coloring problems. Please see the general comment where we discuss the novelty and contribution of our work in more detail.
>
> #### •  **Q2. If the primary contributions are solely related to the architecture and node features, it's worth noting that while the results appear promising, the method might be viewed as somewhat incremental, especially when compared to existing methods like spectral clustering, which can be considered a basic form of unsupervised learning**
>
> We can not agree that contributions might be viewed as somewhat incremental only because they are related to architecture and node features, especially if it contains novelty which allows to significantly outperform existing ones. Generally speaking, the development of a novel unsupervised neural network comes down to the selection of the appropriate architecture, features and loss function. Our work is dedicated to the research in the field of unsupervised graph neural networks, therefore in our experiments we compare QRF-GNN with  1) SOTA unsupervised GNNs, 2) SOTA non-learning based conventional heuristic methods that provide the best solutions on the considered benchmark datasets of the MAX-CUT and graph coloring problems. It would also be interesting for us to compare QRF-GNN with any other unsupervised learning method. However, to the best of our knowledge, there is no a successful application of the spectral clustering method which demonstrates high quality results solving CO problems such as MAX-CUT and coloring.
>
>
>
> **In such a case, could you provide insights into any theoretical guarantees, if any exist for your work?**
>
> As well as many other heuristic algorithms for solving hard combinatorial optimization problems, graph neural networks do not provide theoretical guarantees. The performance of the proposed method is evaluated by numerical experiments and comparative analysis with the SOTA methods.
>
> #### •  **Q3. I have concerns about the reproducibility of this work, as it appears the authors haven't made the code base available. Additionally, they mention that the method lacks stability. How do you determine the best run under these circumstances?**
>
> In the paper we mention the stochasticity of our method, but we do not mean that it lacks stability. The results can vary with the different seed initializations, and this is the common behavior of any other heuristic algorithm that contains some randomness. This is the case for all the algorithms that appear in our experiments section. Therefore, we follow the conventional experimental design for heuristic algorithms with a stochastic component. It is common to report the best value out of several runs and use it in the comparison with other algorithms [1-3]. In addition, in section B of the appendix, you can see the distribution of results for d-regular graphs and each graph from Gset ("default" refers to QRF-GNN). Speaking about stability, in contrast, we can say that GRF-GNN is more robust and stable than, for example, PI-GNN [1], since it provides high quality results for all problems with the same set of hyperparameters.
>
> $ \\ $
>
> We hope that our answers will bring more understanding about the use of GNNs in combinatorial optimization and about the contribution of our paper. We will be glad to answer other questions of the reviewer if they have them.
>
> $ \\ $
>
> [1] Martin J. A. Schuetz, J. Kyle Brubaker, and Helmut G. Katzgraber. Combinatorial optimization with physics-inspired graph neural networks. Nature Machine Intelligence, 4(4):367–377, April 2022a.
>
> [2] W. H. Tonshoff J, Ritzert M and G. M.  Graph neural networks for maximum constraint satisfaction. Frontiers in Artificial Intelligence 2021.
>
> [3] Una Benlic and Jin-Kao Hao. Breakout local search for the max-cutproblem. Engineering Applications of Artificial Intelligence, 26(3):1162–1173, 2013.

---

> > ### Comment · Reviewer_vpKg · 2023-11-19
> >
> > I've thoroughly reviewed all the responses and reviews, and I have additional questions:
> >
> > Is it feasible to assess performance by employing different types of GNNs, or even by incorporating deeper GNN layers? This consideration becomes particularly crucial in the context of unsupervised learning, as certain architectures may impart an implicit regularization effect. Exploring the impact of varying GNN types or layer depths could offer valuable insights.
> >
> > I'm keen to gain a deeper understanding of how the hyperparameters are chosen. While I acknowledge that there is a section in the paper addressing hyperparameters, especially in the realm of unsupervised learning, it is conceivable that the results might vary significantly with different learning rates and other hyperparameters. Conducting experiments to explore the influence of various hyperparameter settings, including learning rates, could enhance the robustness and completeness of the findings.

---

> > > ### Author Response · Authors · 2023-11-22
> > > **Response to Reviewer vpKg**
> > >
> > > Dear reviewer vpKG, we appreciate you following the discussion of our work.
> > >
> > > **A1.**
> > > It is possible that different types of architectures may affect the quality of the QRF-GNN algorithm. Unfortunately,  in the case of graph neural networks, simply increasing the depth leads to an exponential convergence of vertex features to similar values, ​​and the loss of a significant amount of meaningful information occurs [1]. This circumstance was the main reason for increasing the number of layers by making them parallel. We have explained this in the section about QRF-GNN method.
> > >
> > > It's also worth noting that the recurrent feature also seems to bring improvement in the case of other architectures. We conducted an ablation study that showed that the recurrent feature makes the largest fundamental contribution, while adding an additional parallel layer to the architecture, as well as increasing the number of weights in the layer, can only slightly affect the result (please see Fig. 4 and Fig. 5 in the Appendix). This significant contribution is explained by the fact that the QUBO based loss function directly depends on the node classes, and utilizing these classes from the previous step turns GNN training into an iterative optimization process, in which the neural network adjusts the assignment of nodes classes  under the influence of the classes of neighbors.
> > >
> > > [1] T. Konstantin Rusch, Michael M. Bronstein, and Siddhartha Mishra. A survey on over-smoothing in graph neural networks. ArXiv, abs/2303.10993, 2023.
> > >
> > > **A2.**
> > > Indeed, it can be useful to study the robustness of the algorithm with respect to hyperparameters. Following the reviewer's suggestion, we conducted a study on how the learning rate affects convergence. We run the default QRF-GNN architecture with two parallel layers on graphs from the Gset dataset for the Max-Cut problem. As can be seen from Fig. 3 in the updated Appendix, small values ​​of the learning rate do not allow to achieve convergence in $10^5$ iterations. For the learning rate greater than 0.01 the results become relatively stable. and the best cuts ​​are achieved at a rate from 0.01 to 0.02.
> > >
> > > However, we would like to highlight that in this work we did not aim to put additional emphasis on the hyperparameter selection. For the paper results we purposely ran our algorithm on all graphs with the same set of hyperparameters, and even for a different tasks (i.e. Max-Cut and coloring) we only changed the size of the hidden layer, while the rest of the hyperparameters remained the same. The  learning rate used was initially selected out of 3 values  ​​and 4 different hidden layer sizes were tested. A more careful selection of hyperparameters can only improve the result of our algorithm.
> > >
> > > We are happy to answer other questions if they arise

---

> ### Comment · Reviewer_vpKg · 2023-11-22
>
> Thank you for the additional experiments and comments. I have raised the score accordingly. While the method works, I don't think the method is particularly novel, which is why the score is not higher. I am not from this field so I was hoping that some of the other reviewers would advocate for this method.

---

### Official Review · Reviewer_Nnir · 2023-10-31

**Soundness:** 2 fair
**Presentation:** 2 fair
**Contribution:** 2 fair
**Rating:** 6
**Confidence:** 2

**Summary:**

The authors introduce QRF-GNN to efficiently solve problems that have a quadratic unconstrained binary optimization (QUBO) formulation.
The relaxed QUBO objective can be used to perform unsupervised learning for the GNNs.

Further the GNN can be applied repeatedly by including the output probabilities from the previous round as node features in the next round.

**Strengths:**

The method is simple - relax the QUBO formulation with probability predictions and then use these to augment the node features in the next GNN step. Finally repeat until convergence.

For the loss simply minimize (or maximize) the QUBO formulation itself wrt the parameters.

To the best of my knowledge, this is the first time a QUBO probelm has been solved like this.

The results are competitive against other benchmarks algorithms to solve max cut and coloring problems.

**Weaknesses:**

Minor nit - The authors could have elaborated a little on the QUBO formulation and the training details before section 3.

**Questions:**

Did you'll try to solve QUBO problems other than max cut and coloring?

---

> ### Author Response · Authors · 2023-11-15
> **Response to Reviewer Nnir**
>
> We thank the reviewer Nnir for reading our work and considering the results of it worthy of attention.
>
> #### •  **W1.Minor nit - The authors could have elaborated a little on the QUBO formulation and the training details before section 3.**
>
> Unfortunately, the page limit does not allow for much detail to be included in the main body of the article. We have tried to provide a sufficient number of references to the literature so that the interested reader can gain a good understanding of the issues discussed. We hope that the reader will find sufficient technical details placed in the section with experiments and in the appendix, and we are ready to answer any questions regarding the  reproducibility of the results.
>
> #### •  **Q1.Did you'll try to solve QUBO problems other than max cut and coloring?**
>
> We have tried to solve other problems and found the results promising, but the work is not yet finished and this is the subject of further research.

---

> ### Author Response · Authors · 2023-11-22
> **Updates in the paper**
>
> Dear reviewer Nnir,
>
> Following your recommendations, we have added additional explanations of the QUBO formulation to the introduction. Please let us know if you have any further questions or suggestions.
>
> We greatly appreciate your time and effort.
>
> Sincerely, the authors

---

> ### Comment · Reviewer_Nnir · 2023-11-22
> **Response**
>
> I thank the authors for their response and elaborating on QUBO in the paper. I retain my score

---

### Official Review · Reviewer_VZaD · 2023-10-31

**Soundness:** 2 fair
**Presentation:** 2 fair
**Contribution:** 2 fair
**Rating:** 3
**Confidence:** 4

**Summary:**

This paper introduces an unsupervised QRF-GNN method for solving CO problems, characterized by two fundamental attributes: the general quadratic unconstrained binary optimization (QUBO) formulation and its recurrent design. Built upon the QUBO formulation, the proposed method compromises various CO problems including the maximum cut problem and graph coloring problem. Compared to the previous baseline PI-GNN, the main improvements lie in the refinement of the message-passing paradigm within graph neural networks, including introducing the recurrent feature, artificial input features, and parallel graph convolutional layers. Empirical results show that the proposed method outperforms existing unsupervised learning methods and is competitive with conventional heuristics.

**Strengths:**

1.The empirical results show advantages over previous methods built on QUBO formulation and are competitive with conventional heuristics.

2.The framework provides a general solution to a range of CO problems, though this property is derived from the existing QUBO formulation.

**Weaknesses:**

1.The novelty appears to be somewhat limited. It builds upon the foundation laid by PI-GNN [1], inheriting similarities in terms of problem formulation and experimental design. The primary enhancements are centered around the refinement of the message-passing mechanism within the graph neural networks. Moreover, the key characteristics of the proposed QRF-GNN, i.e., the unsupervised approach based on QUBO and the recurrent design, have previously been explored in methods like PI-GNN and RUN-CSP [2]. The methodological innovations do not appear compelling enough to warrant my vote for acceptance.

2.The rationale behind certain design choices remains unclear. Questions arise regarding the decision to incorporate the recurrent feature, artificial input features, and parallel graph convolutional layers in lieu of the raw GNN. If the objective is to enhance representation capacity, why not consider a more advanced GNN or GraphFormer design? It is also necessary to establish the relationship between these design choices and their relevance to CO problems.

3.The evaluation presented in this paper does not convincingly demonstrate QRF-GNN's superiority over heuristic methods. In PI-GNN, Fig. 4 and 5 show its advantages in scalability and computational complexity. Yet these properties are not well verified in this paper.

[1] Combinatorial Optimization with Physics-Inspired Graph Neural Networks. Nature Machine Intelligence 2022.

[2] Graph neural networks for maximum constraint satisfaction. Frontiers in Artificial Intelligence 2021.

**Questions:**

1.Why are recurrent features important for solving CO problems?

2.Since the methodology mainly focuses on the design of the graph networks, have you tried other advanced GNNs or GraphFormers in hand for better performance of this formulation?

3.In Table. 5, why is QRF-GNN more efficient than PI-GNN [1] while the main algorithm pipelines seem similar? It probably comes from the setting of PI-GNN in experiments: "In order to obtain the results of PI-GNN, the authors applied hyperparameter optimization for each graph." However, in my understanding, hyperparameter optimization is not a mandatory request of PI-GNN. It is worth investigating the performance of PI-GNN under the same conditions as QRF-GNN to determine if there is still a speed advantage.

4.In Section 5, the statement mentions, "...considering that most of the mentioned methods are specialized for solving specific problems...". In fact, many methods try to propose a general solving framework for solving broad CO problems, such as [2] [3] [4]. Additionally, solving algorithms for NP-complete (NPC) problems inherently possess the potential to solve other CO problems since any NP problem can be reduced to an NPC problem.

[1] Combinatorial Optimization with Physics-Inspired Graph Neural Networks. Nature Machine Intelligence 2022.

[2] Revisiting Sampling for Combinatorial Optimization. ICML 2023.

[3] DIFUSCO: Graph-based Diffusion Solvers for Combinatorial Optimization. NeurIPS 2023.

[4] From Distribution Learning in Training to Gradient Search in Testing for Combinatorial Optimization. NeurIPS 2023.

**Details Of Ethics Concerns:**

No ethics concerns.

---

> ### Author Response · Authors · 2023-11-15
> **Response to Reviewer VZaD, Part 1.**
>
> We are grateful to the reviewer for carefully studying not only our article, but also related works in the field.
>
>
> #### • **W1. The key characteristics of the proposed QRF-GNN, i.e., the unsupervised approach based on QUBO and the recurrent design, have previously been explored in methods like PI-GNN and RUN-CSP [3].    Q1.Why are recurrent features important for solving CO problems?**
>
> The authors of PI-GNN indeed proposed to use QUBO as a loss function, but their results without hyperparameter optimization are worse than solutions of greedy based algorithms with linear time complexity, as shown by in [1][2]. Optimizing the hyperparameters of GNN for each individual example is quite time-consuming and still does not allow the original PI-GNN to achieve results close to QRF-GNN.
>
> The RUN-CSP algorithm exploits recurrence, but it is different from the one  proposed in our paper. The RUN-CSP uses  the hidden state to update the current state of a node, but the message-passing protocol does not incorporate hidden states of neighbors. It is assumed that the desired information from neighbors of a node was aggregated by the neural network from their hidden states to their current states and thus contributing to the update of the current state of the considered node. As we see from the results in the experiments section, this approach does not yield good quality and the ability of the neural network to aggregate the necessary information is limited.
>
> A key feature of our approach is that we use already predicted nodes classes as hidden recurrent states and then  pass them directly through the message-passing protocol as additional vertex properties. The intuition behind this is as follows. Imagine a Max-Cut task and all vertices at the current step belong to one of the classes. The GNN processes the current vertex and finds that it has the same class as its neighbors. In this case GNN forces a vertex to change its class in order to cut edges with the neighbors and improve the loss value. To summarize, the proposed recurrence is important for solving CO problems, because states of vertices depend on states of nearest neighbors. We will add more details to the article to highlight this point. We have also pointed out this issue in our general comment.
>
>
> #### •  **W2. The rationale behind certain design choices remains unclear. Questions arise regarding the decision to incorporate the recurrent feature, artificial input features, and parallel graph convolutional layers in lieu of the raw GNN.**
>
> We have tried to answer the question about the importance of recurrence above. Specific features and architecture are probably not optimal and indeed are intended to enhance representation capacity without oversmoothing issues. We tried different combinations and, as can be seen from the ablation study in the appendix, recurrence makes the fundamental contribution, while other modifications are incremental. We also tried different convolutional layers such as GCN and GAT, which showed no additional benefits, so we have reported the results for GraphSAGE.
>
> #### •  **Q2. Since the methodology mainly focuses on the design of the graph networks, have you tried other advanced GNNs or GraphFormers in hand for better performance of this formulation?**
>
> It is possible that more modern architectures like GraphFormers are able to improve expressive power, but we believe that the recurrent feature should make a significant contribution in this case as well, as it changes the behavior of the iteration process. This could be a subject for future research, but the computational complexity of training must also be taken into account.
>
> $ \\ $
>
> [1] S. Boettcher, Inability of a graph neural network heuristic to outperform greedy algorithms in solving combinatorial optimization problems, Nature Machine Intelligence 5, 24 (2023)
>
> [2] M. C. Angelini and F. Ricci-Tersenghi, Modern graph neural networks do worse than classical greedy algorithms in solving combinatorial optimization problems like maximum independent set, Nature Machine Intelligence 5, 29 (2023).
>
> [3] W. H. Tonshoff J, Ritzert M and G. M.  Graph neural networks for maximum constraint satisfaction. Frontiers in Artificial Intelligence 2021.

---

> ### Author Response · Authors · 2023-11-15
> **Response to Reviewer VZaD, Part 2.**
>
> #### •  **Q3. In Table. 5, why is QRF-GNN more efficient than PI-GNN [4] while the main algorithm pipelines seem similar? It probably comes from the setting of PI-GNN in experiments: "In order to obtain the results of PI-GNN, the authors applied hyperparameter optimization for each graph." However, in my understanding, hyperparameter optimization is not a mandatory request of PI-GNN. It is worth investigating the performance of PI-GNN under the same conditions as QRF-GNN to determine if there is still a speed advantage.**
>
> Indeed, the difference in runtime arises mainly due to per instance hyperparameters optimization of PI-GNN. It is not mandatory to perform such the optimization. However, the quality of PI-GNN results without optimization may suffer greatly. There are cases where even greedy algorithms perform better [1][2]. We have investigated performance of PI-GNN without per instance optimization also for the coloring problem. We ran the code of PI-GNN presented by the authors on COLOR dataset and citation graphs. We set the hyperparameters in two ways: similar to the  hyperparameters of QRF-GNN (PI-GNN 1) and as an average of reported optimal values obtained during optimization (PI-GNN 2). The averaging was carried out separately for the COLOR dataset and citations graphs since their optimal parameters vary greatly. The results in columns "QRF-GNN" and "PI-GNN" of the following table refer to the Table 3 in the paper. The results in "PI-GNN 1"  and "PI-GNN 2" columns are taken under the same conditions as QRF-GNN (the best value out of 10 runs is presented).
>
> |Graph|QRF-GNN|PI-GNN|PI-GNN 1|PI-GNN 2|
> |-----|-------|------|--------|--------|
> |anna| 0 | 0 | 1 | 1 |
> |jean| 0 | 0 | 0 | 1 |
> |myciel5| 0 | 0 | 0 | 0 |
> |myciel6| 0 | 0 | 0 | 0 |
> |queen5_5| 0 | 0 | 0 | 4 |
> |queen6_6| 0 | 0 | 3 | 1 |
> |queen7_7| 0 | 0 | 9 | 8 |
> |queen8_8| 0 | 1 | 4 | 1 |
> |queen9_9| 0 | 1 | 6 | 2 |
> |queen8_12| 0 | 0 | 1 | 2 |
> |queen11_11| 7 | 17 | 19 | 15 |
> |queen13_13| 15 | 26 | 31 | 27 |
> |Cora| 0 | 0 | 0 | 90 |
> |Citeseer| 0 | 0 | 0 | 50 |
> |Pubmed| 0 | 17 | 3 | 1250 |
>
> As can be seen from the table, there is a noticeable drop in overall quality. The algorithm fails to find a solution without violations for graphs, which were successfully colored by using per instance optimization. Therefore we decided to consider the optimization process as part of the PI-GNN algorithm. In order to keep the results of PI-GNN competitive, we place the runtime reported by the authors in [3] in Table 5. The QRF-GNN method does not require optimization of hyperparameters for each instance to obtain accurate results, therefore the table shows its runtime without optimization. Perhaps it is possible to find general hyperparameters that allow PI-GNN to work properly, but the authors of [3] do not provide them and the corresponding runtimes, which forces us to compare with what is given in [3].
>
> $ \\ $
>
> [1] S. Boettcher, Inability of a graph neural network heuristic to outperform greedy algorithms in solving combinatorial optimization problems, Nature Machine Intelligence 5, 24 (2023)
>
> [2] M. C. Angelini and F. Ricci-Tersenghi, Modern graph neural networks do worse than classical greedy algorithms in solving combinatorial optimization problems like maximum independent set, Nature Machine Intelligence 5, 29 (2023).
>
> [3] M. J. A. Schuetz, J. K. Brubaker, Z. Zhu, and H. G. Katzgraber, Graph coloring with physics-inspired graph neural networks, Phys. Rev. Res. 4, 043131 (2022)
>
> [4] Martin J. A. Schuetz, J. Kyle Brubaker, and Helmut G. Katzgraber. Combinatorial optimization with physics-inspired graph neural networks. Nature Machine Intelligence, 4(4):367–377, April 2022a.

---

> ### Author Response · Authors · 2023-11-15
> **Response to Reviewer VZaD, Part 3.**
>
> #### •  **W3.The evaluation presented in this paper does not convincingly demonstrate QRF-GNN's superiority over heuristic methods.**
>
> In our experiments we   focused mainly on the comparison between QRF-GNN and existing unsupervised GNNs. As for the heuristics, our intention was to put QRF-GNN in the context of the algorithms that provide the best solutions on the considered benchmark datasets (BLS and TSHEA for Gset, HybridEA for COLOR). In fact, there are dozens of heuristics against which QRF-GNN does demonstrate superiority  (e.g. the heuristics presented in Table 1 in [4] , Table 2 in [5] and Table 1 in [3]). However, we present the comparison with the state-of-the-art methods and show that our algorithm is able to outperform them on some instances. In addition, we also compare our GNN with the strong EO heuristic for Max-Cut, which is able to achieve almost optimal result for d-regular graphs [1], and show that QRF-GNN outperforms it.
> We agree that QRF-GNN does not show the best results among all existing methods on the considered benchmark instances. But we would like to question whether this is a weakness taking into account the ability to solve a wide range of CO problems, the linear scalability and the large gap between the results of best conventional heuristics and previously known unsupervised learning methods. Please see also the general comment explaining the results.
>
> **In PI-GNN, Fig. 4 and 5 show its advantages in scalability and computational complexity. Yet these properties are not well verified in this paper.**
>
> We investigated scalability and computational complexity for large random d-regular graphs with vertex counts ranging from 50 thousand to 1 million and found dependence similar to that in PI-GNN (please see Fig. 2 in the appendix). Unlike in Fig. 4 in [4], we calculated the P-value, since the plot scale for the number of cuts does not allow to see the difference in quality for graphs with different numbers of vertices. Additionally, we raised the question of convergence, which has not been done for PI-GNN. Despite the fact that there were not enough iterations for convergence on large graphs, the P-value was still competitive (please compare Fig. 1 in [1] and Fig. 2b in the appendix) given the near-linear computational complexity. It should also be noted that Figs. 4 and 5 in  [4] as well as Fig. 2 in the appendix refer to the scalability of the algorithms without hyperparameter optimization. Under such conditions PI-GNN performs worse then the simple greedy search, while QRF-GNN provides results close to the global optimal bounds (see Fig.1 in [1] and the answer to Q3).
>
> $ \\ $
>
> [1] S. Boettcher, Inability of a graph neural network heuristic to outperform greedy algorithms in solving combinatorial optimization problems, Nature Machine Intelligence 5, 24 (2023)
>
> [2] M. C. Angelini and F. Ricci-Tersenghi, Modern graph neural networks do worse than classical greedy algorithms in solving combinatorial optimization problems like maximum independent set, Nature Machine Intelligence 5, 29 (2023).
>
> [3] M. J. A. Schuetz, J. K. Brubaker, Z. Zhu, and H. G. Katzgraber, Graph coloring with physics-inspired graph neural networks, Phys. Rev. Res. 4, 043131 (2022)
>
> [4] Martin J. A. Schuetz, J. Kyle Brubaker, and Helmut G. Katzgraber. Combinatorial optimization with physics-inspired graph neural networks. Nature Machine Intelligence, 4(4):367–377, April 2022a.
>
> [5] W. H. Tonshoff J, Ritzert M and G. M.  Graph neural networks for maximum constraint satisfaction. Frontiers in Artificial Intelligence 2021.

---

> ### Author Response · Authors · 2023-11-15
> **Response to Reviewer VZaD, Part 4.**
>
> #### •  **Q4. In Section 5, the statement mentions, "...considering that most of the mentioned methods are specialized for solving specific problems...".**
>
> In the text of our paper, the quote was referring to the heuristics mentioned in the previous sentence: BLS and TSHEA for Max-Cut and HybridEA for graph coloring. Our aim was to highlight that general GNN solver can compete with the best specialized algorithms.
>
> **In fact, many methods try to propose a general solving framework for solving broad CO problems, such as [1] [2] [3].**
>
> The algorithms mentioned by the reviewer are very interesting for study and practical use. However, the methods [2],[3] are supervised, which means that the training set of already labeled (i.e. solved) problems is required. Although the supervised learning framework for solving CO problems is very popular, these methods cannot be considered as autonomous algorithms and the comparison between supervised and unsupervised approaches is methodologically incorrect. The method [1] does not need a training set, and we would be interested in comparing QRF-GNN with this method. However, it is not learning-based algorithm, and in our comparative analysis we mainly focused on unsupervised GNNs.
>
> **Additionally, solving algorithms for NP-complete (NPC) problems inherently possess the potential to solve other CO problems since any NP problem can be reduced to an NPC problem.**
>
> We doubt the practical application of this statement and have not seen any algorithm based on this in the literature. We would like to ask the reviewer to share examples if they know some. Sometimes it makes sense to formulate one problem in terms of another. For instance, Mixed-Integer Linear Programming  and QUBO formulations have proven its effectiveness in practice. However, this is not directly related to polynomial-time reductions in NP-completeness theory.
>
> $ \\ $
>
> [1] Haoran Sun,  Katayoon Goshvadi, Azade Nova, Dale Schuurmans, Hanjun Dai. Revisiting Sampling for Combinatorial Optimization. ICML 2023.
>
> [2] Zhiqing Sun, Yiming Yang. DIFUSCO: Graph-based Diffusion Solvers for Combinatorial Optimization. NeurIPS 2023.
>
> [3] Yang Li, Jinpei Guo, Runzhong Wang, Junchi Yan. From Distribution Learning in Training to Gradient Search in Testing for Combinatorial Optimization. NeurIPS 2023.

---

> ### Author Response · Authors · 2023-11-22
> **to Reviewer VZaD**
>
> Dear Reviewer VZaD,
>
> Given that time is very short, we look forward to your feedback. We hope that our answers and explanations were convincing. Please let us know if we were able to answer your questions.
>
> We greatly appreciate your time and effort.
>
> Sincerely, the authors

---

> ### Comment · Reviewer_VZaD · 2023-11-23
> **Thanks for the response**
>
> Thanks for the reply. I appreciate the effort you've put into this. However, I have remaining concerns.
>
> 1. The method is still not novel to me. I understand that QRF-GNN maintains some differences from PI-GNN and RUN-CSP, but it still seems to be a close effort with similar designs, since RUN-CSP has already considered the recurrent features. If your difference to RUN-CSP lies in some specific message-passing design like the use of hidden states of neighbors, you should take this as your core method novelty to explain. However, such novelty may still not be enough for me.
>
> 2. How are the scalability and computational complexity of QRF-GNN compared to other baselines?

---

> ### Author Response · Authors · 2023-11-23
> **Response to Reviewer VZaD**
>
> **A1.** We apologize that our previous explanations and experiments have not emphasized novelty to the extent necessary to influence your decision. In conclusion, we would like to highlight some points. The RUN-CSP work is not a pioneer article in the study of recurrence for graph neural networks. Attempts to apply recurrence have existed since at least 2017, when the authors of GraphSAGE proposed using the LSTM aggregation function [1]. In this sense, the RUN-CSP architecture is more similar to the GraphSAGE with LSTM aggregation than to QRF-GNN. However, previous recurrent architectures failed to produce high quality solutions for combinatorial problems. We propose a different type of recurrence, which we described in detail in our previous answers (Part 1) and in Section 3, in particular in the scheme of Algorithm 1. The recurrence mechanism is generally a widely used technique in deep learning, but can take a variety of significantly different forms that can fundamentally change the behavior of the entire architecture. In addition, the difference between RUN-CSP and QRF-GNN is not only in the different implementation of recurrence, but also in the problem formulation, the loss function used, the method of aggregation of neighboring states, and the method of training (RUN-CSP is additionally pretrained on synthetic data in order to be further trained on other instances). We still believe that the results obtained using our architecture deserve the attention of the community, since they are qualitatively superior to the results from other unsupervised graph neural networks.
>
> $
> \\
> $
>
> **A2.** The main baselines in our paper are unsupervised graph neural networks, in particular PI-GNN and RUN-CSP. As we have written above, in the RUN-CSP paper the authors pretrain the model on synthetic instances, so it is challenging to make a fair comparison with it. As another baseline, we present the results of the PI-GNN with per-instance hyperparameter optimization. As we discussed in parts 2,3 of our previous response as well as in the section 4 of the paper, the authors of the algorithm give scalability for the algorithm without hyperparameter optimization, which obtains results of very low quality. If compared with the setting of the PI-GNN, whose results are used as a baseline, we have done it terms of computational time (Table 5.) and demonstrated the advantage.
>
> By reporting Fig. 2 we aimed to show the scalability and linear ~$O(n)$ complexity of QRF-GNN on d-regular graphs, which is optimal in the context of approximate solution methods for combinatorial optimization problems. Hence, we considered this result to be self-sufficient, and it allows the authors of other papers to compare with our algorithm on this criterion. For d-regular graphs we also compare with non-learning EO [2] heuristic baseline which has the theoretical complexity of $O(n^4)$.
>
> $
> \\
> $
>
> [1] William L. Hamilton, Rex Ying, and Jure Leskovec. Inductive representation learning on large graphs. In Proceedings of the 31st International Conference on Neural Information Processing Systems, NIPS’17, pp. 1025–1035, Red Hook, NY, USA, 2017b. Curran Associates Inc. ISBN 9781510860964.
>
> [2]Stefan Boettcher and Allon G. Percus. Optimization with extremal dynamics. Physical Review Letters, 86(23):5211–5214, June 2001. doi: 10.1103/physrevlett.86.5211.
>
> $
> \\
> $
>
> We hope that we have been able to address the reviewer's concerns.
>
> Best regards, authors.

---

### Official Review · Reviewer_oXir · 2023-11-08

**Soundness:** 2 fair
**Presentation:** 3 good
**Contribution:** 2 fair
**Rating:** 3
**Confidence:** 4

**Summary:**

This paper leverages the GNNs to solve the quadratic unconstrained binary optimization problems, especially for the maximum cut and graph coloring problems.  Experimental results demonstrate the effectiveness of the proposed method.

**Strengths:**

1. Using Neural Networks to solve combinatorial optimization problems is an interesting topic.
2. The proposed QRF-GNN can outperform existing GNNs in solving Max-Cut and graph coloring problems.
3. The paper is well-written and easy to follow.

**Weaknesses:**

1. The paper's novelty appears constrained, utilizing a GNN with QUBO's continuous relaxation as a loss function for combinatorial optimization. Clarification on how this differs from prior work would be beneficial.

2. The motivation behind the specific design of the GNN, particularly the use of two different convolutional layers (SAGEConv Pool and SAGEConv mean) in the first layer, is not explained. A rationale for these choices should be provided.

3. An ablation study is needed to substantiate the proposed GNN's superiority over existing methods. Without this, the advantage of the proposed model remains unclear.

**Questions:**

Please refer to the weaknesses.

---

> ### Author Response · Authors · 2023-11-15
> **Response to Reviewer oXir**
>
> We thank the reviewer oXir for their comment.
>
>
> #### •  **Q1. The paper's novelty appears constrained, utilizing a GNN with QUBO's continuous relaxation as a loss function for combinatorial optimization. Clarification on how this differs from prior work would be beneficial.**
> The main difference from the previous architectures is the use of recurrent node features within the message-passing protocol. We have written a more detailed answer in the general comment, please find additional information there.
>
> #### •  **Q2. The motivation behind the specific design of the GNN, particularly the use of two different convolutional layers (SAGEConv Pool and SAGEConv mean) in the first layer, is not explained. A rationale for these choices should be provided.**
> The use of different parallel layers was intended to improve representation capacity while preventing the oversmoothing effect. It should be noted, that the particular design of the architecture plays minor role, since the main gain in quality comes from the recurrence, which is supported by the ablation study in the section B of the appendix.
>
>
> #### •  **Q3. An ablation study is needed to substantiate the proposed GNN's superiority over existing methods. Without this, the advantage of the proposed model remains unclear.**
> In the appendix of the paper we provide a detailed ablation study of our method, showing the contribution of each part of the algorithm.  To demonstrate the advantage of the proposed GNN over existing methods, numerical experiments are given in section 4 of the main body of the paper. Could you please give us more specific request, what other analyses would be helpful for us to illustrate the advantage of the proposed model?

---

> ### Author Response · Authors · 2023-11-22
> **to Reviewer oXir**
>
> Dear Reviewer oXir,
>
> Given that time is very short, we look forward to your feedback. Please also let us know if you have any additional questions or concerns about our work.
>
> We greatly appreciate your time and effort.
>
> Sincerely, the authors

---

> > ### Comment · Reviewer_oXir · 2023-11-22
> >
> > Thanks for the response from the authors. I still have questions about the novelty and motivation of the proposed designs. It seems the novelty lies in the use of recurrent node features. However, I didn't see much emphasis on the recurrent node feature in the paper, and the motivation for using recurrent node features is unclear to me. Also, authors said, "The use of different parallel layers was intended to improve representation capacity while preventing the oversmoothing effect"; "Other modifications of the architecture (i.e. parallel convolutional layers) and different static features were used to increase the expressive power of GNN. " Where do these conclusions come from?

---

> > > ### Author Response · Authors · 2023-11-22
> > > **Response to Reviewer oXir**
> > >
> > > #### **Q. Thanks for the response from the authors. I still have questions about the novelty and motivation of the proposed designs. It seems the novelty lies in the use of recurrent node features. However, I didn't see much emphasis on the recurrent node feature in the paper, and the motivation for using recurrent node features is unclear to me.**
> > >
> > > Dear reviewer, please allow us to further outline the motivation for using the recurrent feature. The QUBO based loss function is directly determined by the state of vertices relative to the state of their neighbors. For example, in the graph coloring problem, if two adjacent vertices have the same class, i.e. have the same color, then the value of the loss function will proportionally increase by 1. The optimal solution is reached when all neighboring vertices have different colors, in which case the loss function value is zero. The information about what colors are assigned to the neighbors of the current vertex allows the neural network to change the color of this vertex accordingly. To the best of our knowledge, current vertex classes have not previously been used recursively within the message-passing protocol.
> > >
> > >
> > > If the recurrent feature is removed from the architecture, the quality of the remaining model's results will be comparable to those of simple greedy algorithms. Using only the recurrent feature without one of the parallel layers will already allow to surpass the results of other unsupervised GNNs. After incorporating parallel layers and predefining specific features for nodes (e.g. random vectors, pagerank, and shared vectors), the results on the considered problems can compete with the best specialized heuristics and may even exceed them on certain graphs. All of the these statements are supported by the ablation study in the Appendix.
> > >
> > > In Section 3, we describe the mechanism of the recurrent feature as part of QRG-GNN, and in Sections 1,3,5 we try to emphasize its importance, in particular by drawing on numerical experiments and the ablation study.
> > >
> > > $
> > > \\
> > > $
> > >
> > >
> > > #### **Also, authors said, "The use of different parallel layers was intended to improve representation capacity while preventing the oversmoothing effect"; "Other modifications of the architecture (i.e. parallel convolutional layers) and different static features were used to increase the expressive power of GNN. " Where do these conclusions come from?**
> > >
> > > Different aggregation functions in parallel layers allow to accumulate information of different properties. As the ablation study shows, in some cases it allows to get an advantage over using one layer with one fixed aggregation function, even if we double the number of parameters in this layer. Unfortunately, increasing the depth in graph neural networks is challenging, since it leads to an exponential convergence of vertex features to similar values, and a significant amount of meaningful information is lost. This over-smoothing issue is a known problem in the community [1-3].
> > >
> > > $
> > > \\
> > > $
> > >
> > > [1] T. Konstantin Rusch, Michael M. Bronstein, and Siddhartha Mishra. A survey on over-smoothing in graph neural networks. ArXiv, abs/2303.10993, 2023.
> > >
> > > [2] Oono, K. (2019, May 27). Graph Neural Networks Exponentially Lose Expressive Power for Node Classification. ArXiv.Org. https://arxiv.org/abs/1905.10947
> > >
> > >  [3] Chen, D. (2019, September 7), Measuring and Relieving the Over-smoothing Problem for Graph Neural Networks from the Topological View. ArXiv.Org. https://arxiv.org/abs/1909.03211
> > >
> > > $
> > > \\
> > > $
> > >
> > > We also refer you to our responses to the questions raised by the VZaD reviewer, as some of them are closely related to yours.
> > >
> > > We will be glad to answer additional questions from the reviewer if they arise.

---

### Author Response · Authors · 2023-11-15
**General Comment**

Dear reviewers, thank you all for reading the paper and providing valuable comments which will help us to improve the quality of the manuscript. Unfortunately, it seems to us that the novelty and main contribution of our work remained unclear to most reviewers. Therefore, we decided to write a general comment with a detailed explanation of these issues.
$ \\ $
### **•The algorithmic novelty.**
The main contribution of our method is to use nodes classes predicted on the previous step as hidden recurrent states, and then directly utilize them through the message-passing protocol as an additional vertex feature. This distinguishes our architecture from all others and allows the graph neural network to collect information about the state of the neighbors for each vertex in the most efficient way. This information allows to make optimal decision on changing the current state of each considered vertex. Thus, the entire GNN training can be viewed as an iterative optimization process in which vertices change classes depending on the classes of neighbors and their initial properties.
Other modifications of the architecture (i.e. parallel convolutional layers) and different static features were used to increase the expressive power of GNN. They only incrementally improve the result of the vanilla GNN with added recurrence (please refer to the ablation study in the appendix) and can be changed. We will highlight this point in the updated version of our paper.

$ \\ $

### **•The results**
The use of the recurrent features together with additional modifications of the GNN architecture allows our method to significantly outperform all other unsupervised GNN architectures in solving the considered CO problems. Until now, the question of whether unsupervised GNNs can obtain accurate solutions to hard CO problems has been open, and their main advantage has been considered to be scalability and the ability to solve large problems. We refer here to a recent discussion about the performance of PI-GNN [1-5]. We believe that the QRF-GNN results may encourage the community to rethink these issues. We show that unsupervised GNNs can not only outperform advanced heuristics such as EO, but even be comparable to the SOTA heuristics specilized for solving MAX-CUT and coloring problems, and defeat them in some cases. To the best of our knowledge, no unsupervised learning method could achieve this before. Therefore, we consider our results significant for the emerging ML4CO (Machine Learning for Combinatorial Optimization) field.

$ \\ $

[1] Martin J. A. Schuetz, J. Kyle Brubaker, and Helmut G. Katzgraber. Combinatorial optimization with physics-inspired graph neural networks. Nature Machine Intelligence, 4(4):367–377, April 2022a.

[2] S. Boettcher, Inability of a graph neural network heuristic to outperform greedy algorithms in solving combinatorial optimization problems, Nature Machine Intelligence 5, 24 (2023)

[3] M. C. Angelini and F. Ricci-Tersenghi, Modern graph neural networks do worse than classical greedy algorithms in solving combinatorial optimization problems like maximum independent set, Nature Machine Intelligence 5, 29 (2023).

[4] Martin J. A. Schuetz, J. Kyle Brubaker, and Helmut G. Katzgraber. Reply to: Inability of a graph neural network heuristic to outperform greedy algorithms in solving combinatorial optimization problems. Nature Machine Intelligence, 5(1):26–28, January 2023

[5] Martin JA Schuetz, J Kyle Brubaker, and Helmut G Katzgraber. Reply to: Modern graph neural networks do worse than classical greedy algorithms in solving combinatorial optimization problems like maximum independent set. Nature Machine Intelligence, 5(1):32–34, 2023.

---

### Author Response · Authors · 2023-11-22
**Updated paper submitted**

Dear reviewers, we have updated our paper according to the reviews.

1) Following the recommendation of reviewer Nnir, we have added more explanations on QUBO problem in the introduction (p.1, eq. 1).

2) In response to reviewer vpKg's question, we have added experiments in section A. of the appendix showing the dependence of the algorithm's performance on the chosen learning rate (p. 14, fig. 3).

3) Explanations on the choice of network architecture and the importance of the recurrent feature have been highlighted in section 3 (p. 4).

---

### Meta-Review · Area_Chair_dN3i · 2023-12-06

**Metareview:**

This paper introduces QRF-GNN, a new variant of graph neural network (GNN) for solving Quadratic Unconstrained Binary Optimization (QUBO) problems in an unsupervised manner. Specifically, it applies GNN repeatedly and uses the output probabilities from the previous round to augment the node features in the next round. This process repeats until convergence.  The empirical results on max cut and graph coloring problems show the advantage of QRF-GNN over previous methods built on QUBO formulation and to be competitive with conventional heuristics.

The reviewers  (oXir, VZaD, vpKg) recognize that the problem studied in this paper is interesting and fundamental, and that the empirical results are encouraging.  However, the limited novelty of this paper is a major concern (oXir, VZaD, vpKg) as both the problem foundation (including the unsupervised nature and the recurrent design) and experimental design in this paper are similar to those already explored by previous methods.  Also, the motivation/rationale for certain specific design choices (regarding the incorporation of recurrent features, artificial input features, parallel graph convolutional layers etc.) is not made sufficiently clear.  The authors tried to debate some of the above concerns but failed to convince the reviewers fully.

A rejection is recommended.

**Justification For Why Not Higher Score:**

This paper lacks novelty and the motivation for the proposed method is not clearly presented in the paper.

**Justification For Why Not Lower Score:**

N/A

---

### Decision · Program_Chairs · 2024-01-16

Reject